# Comparative Transcriptomics Provides Insight into Floral Color Polymorphism in a *Pleione limprichtii* Orchid Population

**DOI:** 10.3390/ijms21010247

**Published:** 2019-12-30

**Authors:** Yiyi Zhang, Tinghong Zhou, Zhongwu Dai, Xiaoyu Dai, Wei Li, Mengxia Cao, Chengru Li, Wen-Chieh Tsai, Xiaoqian Wu, Junwen Zhai, Zhongjian Liu, Shasha Wu

**Affiliations:** 1Key Laboratory of National Forestry and Grassland Administration for Orchid Conservation and Utilization, College of Landscape Architecture, Fujian Agriculture and Forestry University, Fuzhou 350002, China; zyy211802@126.com (Y.Z.); daizhongwu1995@163.com (Z.D.); dxy1020239286@126.com (X.D.); liwei961214@163.com (W.L.); mengxiacao0302@126.com (M.C.); lcr5060@126.com (C.L.); tsaiwc@mail.ncku.edu.tw (W.-C.T.); wu_xiaoqian@126.com (X.W.); zhai-jw@163.com (J.Z.); 2Huanglong National Scenic Reserve, Songpan 623300, China; zth_17@tom.com; 3Institute of Tropical Plant Sciences and Microbiology, National Cheng Kung University, Tainan City 701, China

**Keywords:** *Pleione limprichtii*, flower color polymorphism, variation within populations, metabolome analysis, anthocyanin biosynthetic pathway, RNA sequencing, transcription factor

## Abstract

Floral color polymorphism can provide great insight into species evolution from a genetic and ecological standpoint. Color variations between species are often mediated by pollinators and are fixed characteristics, indicating their relevance to adaptive evolution, especially between plants within a single population or between similar species. The orchid genus *Pleione* has a wide variety of flower colors, from violet, rose-purple, pink, to white, but their color formation and its evolutionary mechanism are unclear. Here, we selected the *P. limprichtii* population in Huanglong, Sichuan Province, China, which displayed three color variations: Rose-purple, pink, and white, providing ideal material for exploring color variations with regard to species evolution. We investigated the distribution pattern of the different color morphs. The ratio of rose-purple:pink:white-flowered individuals was close to 6:3:1. We inferred that the distribution pattern may serve as a reproductive strategy to maintain the population size. Metabolome analysis was used to reveal that cyanindin derivatives and delphidin are the main color pigments involved. RNA sequencing was used to characterize anthocyanin biosynthetic pathway-related genes and reveal different color formation pathways and transcription factors in order to identify differentially-expressed genes and explore their relationship with color formation. In addition, qRT-PCR was used to validate the expression patterns of some of the genes. The results show that *PlFLS* serves as a crucial gene that contributes to white color formation and that *PlANS* and *PlUFGT* are related to the accumulation of anthocyanin which is responsible for color intensity, especially in pigmented flowers. Phylogenetic and co-expression analyses also identified a R2R3-MYB gene *PlMYB10*, which is predicted to combine with *PlbHLH20* or *PlbHLH26* along with *PlWD40-1* to form an MBW protein complex (MYB, bHLH, and WDR) that regulates *PlFLS* expression and may serve as a repressor of anthocyanin accumulation-controlled color variations. Our results not only explain the molecular mechanism of color variation in *P. limprichtii*, but also contribute to the exploration of a flower color evolutionary model in *Pleione*, as well as other flowering plants.

## 1. Introduction

Flower color is one of the most attractive characteristics of plants in nature. With such massive variation, flower color is regarded as an evolutionarily labile trait and has been shown to contribute to plant evolution [1,2,3]. In particular, flower color adaptive mutations mediated through pollinators are directly relevant to phenotypic evolution [4]. Color variation is considered a fixed difference between species and promotes the formation of population polymorphism [5,6]. Among angiosperms, sister species always display differences in flower hue and intensity, This pattern of flower color polymorphism is used as a model trait in the study of ecology, evolution, and gene regulation [7]. Color changes are related to flower pigment content. To date, the molecular mechanism of flower color transition has been investigated in several species, owing to the main floral pigments having been well characterized in many plants [8,9,10,11,12,13] providing sufficient information for studying floral color formation in non-model species and the opportunity to explore the relationship between phenotypic evolution and color variations.

Although flower color is influenced by many factors, flavonoids, especially anthocyanins which are produced by the anthocyanin biosynthesis pathway (ABP), are the primary components that contribute to floral pigments and they are produced by highly conservative structural and regulatory components [14]. The ABP involves multi-metabolic processes which mainly consist of seven core structural genes: *CHS*, *CHI*, *F3H*, *F3′H*, *F3′5′H*, *DFR*, and *ANS*, and several branch-enzyme genes [15]. Due to the instability of anthocyanidins, they exist mainly as anthocyanins, which are formed by anthocyanidins and various glycosides [16]. They play an irreplaceable role in the color development of plants and are primarily derived from three main anthocyanidins: Pelargonidin (brick red to scarlet), cyaniding (red to magenta), and delphinidin (purple to violet) [16]. Studies have shown that blocking the ABP can lead directly to variations in pigment production and affect flower color [17]. In addition to the structural genes in the ABP, transcription factors also contribute to flower color transition by regulating the spatial and temporal expression of the structural genes [18,19]. The ABP is regulated by three complex, interacting transcription factors: R2R3-MYB, basic helix–loop–helix (bHLH), and WD40-repeat (WDR) [20]. These transcription factors activate or suppress the transcription and expression of target genes, thereby regulating anthocyanin synthesis [21]. Generally, the structural and regulatory genes involved in the ABP have provided a number of targets to reveal the diversity of mutations that could block the ABP [22]. For flower polymorphism within populations, locating the blockage could elucidate the cause of flower color transition at the biochemical and molecular scales [8,10,11]. In addition, understanding their specific ABP is of benefit for predicting evolutionary influences from a genetic perspective.

The genus *Pleione* (Orchidaceae) comprises nearly 30 species of terrestrial, lithophytic, and epiphytic plants with high ornamental value [23]. There are 27 species in China, while Yunnan is the world biodiversity distribution center of this genus [24,25]. *Pleione* possesses different flower colors ranging between white, pink, lavender, magenta, light purple, and yellow [26]. In particular, populations of Huanglong, Sichuan Province, there remain a color polymorphism population, consisting of pink flowers of different intensities along with white mutant individuals, which can be considered an ideal situation to study the polymorphism formation mechanisms of *Pleione*, as well as benefit to explore potential correlation between color pattern and the species evolution.

The focus of our study was to understand the molecular mechanism of color polymorphism, including how the white individuals formed and the main reason caused pink flowers intensities, as well as summarize the general rules of the color distribution pattern. We aimed to investigate distribution of color monomorphic in the Huanglong *P. limprichtii* population and examine the transcriptome and biochemistry of their color polymorphic petals. RNA sequencing (RNA-seq) and ultra-performance liquid chromatography (UPLC) were used to identify the variation of related genes and the differences in flavonoid intermediates in the ABP that cause color transition, respectively.

## 2. Results

### 2.1. Color Differences and Quantity Distribution Pattern

After applying quantitative statistics to the individuals with the three distinct flowers at the full bloom stage randomly distributed in three rock populations, we concluded that the rose-purple individuals account for nearly 60%, the pink ones occupied about 30%, and the white ones not more than 10% (Table 1). The ratio of the number of the three differently flowered plants (rose-purple:pink:white) is roughly 6:3:1. Thus rose-purple is the main flower color in the Huanglong population, while white is the rarest color.

According to principle component analysis, there were significant differences in the a*(redness and greenness) values of the three colors (Figure 1. A comparison of L* (lightness) and C* (chroma) values among the three color groups indicated that C* values can also be used as an indicator to distinguish these three colors, since the rose-purple group has the highest C* value, the white group the lowest, and the pink group an intermediate value (Table 2). 

### 2.2. Major Anthocyanin Compounds in P. limprichtii

UPLC analysis revealed that four anthocyanins and derivatives were detected in both the pigmented and white flower petals: Cyanidin 3-O-glucosyl-malonylglucoside, cyanidin 3-O-malonylhexoside, cyanidin, and delphinidin (Table 3). The cyanidin accumulation and delphinidin accumulation branches of the ABP in *P. limprichtii* indicate that anthocyanins and their derivatives are the main flower color pigments.

### 2.3. RNA-Seq and Annotation of Unigenes

To understand the molecular basis of flower polymorphism in *P. limprichtii*, three distinctly colored flower petals and lips were used for RNA-seq. A total of 80,525 unigenes with a mean length of 856 bp were obtained by de novo assembly. We assessed the quality of the unigenes in the transcriptome library. The length of N50 (sequence length of the shortest contig at 50%) was 1470 bp, the Q20 and Q30 percentages were 98% and 95%, the GC content was 40%, and the unigenes were generally distributed between 200 bp and 3000 bp in length. A total of 69,004,304 base pairs were aligned. These data show that the throughputs and sequencing quality were high enough to ensure further analysis. 

According to the BLASTx results, a total of 33,724 unigenes were annotated, accounting for 41.88% of all the unigenes. Among these, 33,459 could be annotated using the Nr database (Non-redundant protein database, 41.55%), 21,177 using the Swiss-prot database (Swiss-Protein protein database, 26.30%), 11,067 using the KEGG database (Kyoto encyclopedia of genes and genomes, 24.68%), and 19,871 using the KOG database (Eukaryotic orthologous groups, 13.74%). In addition, there were 8396, 151, 42, and 23 unigenes annotated only using the Nr, Swiss-protein, KEGG, and KOG databases, respectively (Figure 2).

Statistical analysis of the E-value (The probability due to chance) characteristics of the Nr annotations revealed that 32.05% of the mapped sequences showed strong homology (E-value < 1 × 10^-3^), while 29.48% in the Swiss-protein database, 44.77% in the KEGG database, and 30.07% in the KOG database showed strong homology (Figure 3).

Based on the Nr annotations, 6495 unigenes were classified into 53 functional categories, belonging to three functional terms: molecular function, cellular component, and biological process. The largest percentages of unigenes identified within each of the three functional terms were metabolic process (3602 unigenes), cell and cell part (2237 and 2233 unigenes), and binding and catalytic activity (2481 and 3409 unigenes), corresponding to biological process, cellular component, and molecular function, respectively (Figure 4).

To exhaustively explore the potential functions of the annotated unigenes, 11,067 unigenes were mapped onto 131 KEGG pathways, including metabolic pathways, biosynthesis of secondary metabolites, biosynthesis of antibiotics, and many other important metabolic pathways. Metabolic pathways and biosynthesis of secondary metabolites were enriched in the most unigenes, with 2450 and 1294 unigenes annotating to these two pathways, respectively, while the anthocyanin pathway was enriched in only one unigene. Among these 131 pathways, there are two color formation-related pathways, the flavonoid biosynthesis pathway (ko00941) to which 36 unigenes were mapped, and the ABP (ko00942) to which one unigene was mapped (the pathways are listed in Appendix A).

### 2.4. The Expression Patterns of ABP Genes

Genes involved in the flavonoid biosynthesis and ABPs that are related to color formation were isolated, including structural and regulatory genes, in order to identify those involved in the regulation of flower polymorphism in *P. limprichtii*. There were 21 ABP-related structural unigenes, which encoded 10 enzymes, expressed both in pigmented flowers and white flowers, with a total of 14 R2R3-MYB, 32 bHLH, and 23 WD40 regulatory unigenes. With regard to structural unigenes, six of the 10 belong to multigene families, with the exception of *F3H, F3′5′H, ANR*, and *FLS*, which were single copy genes (Table 4). All of these unigenes were used to analyze the expression pattern of the flower color polymorphism of *P. limprichtii*.

Since petal anthocyanins are detectable in different color petals, we inferred that floral color differences were caused by different expression patterns of ABP-related genes. The results show that the genes encoding flavonol synthese (*FLS*, *PlFLS*), anthocyanin synthese (*ANS*, *PlANS1*, *PlANS2*), and UDP-glucose anthocyanidin 3-O-glucosyltransferase (*UFGT*, *PlUFGT1*, *PlUFGT2*) were expressed at a higher level in pigmented flowers (rose-purple and pink) than in white flowers, and were more up-regulated in the rose-purple flowers than in the pink flowers (Figure 5a and Figure 6). These genes were correlated with flower color intensity and color phenotypes. Besides these directly affected genes, other genes that have different expression patterns between pigmented flowers and white flowers also influence color formation, such as the flavanone 3′-hydroxylase gene (*F3′H*; *Pl F3′H3*) and dihydroflavonol 4- reductase (*DFR*, *PlDFR3*). These were both up-regulated in rose-purple flowers and contribute to red color formation. According to the ABP-related gene expression patterns and metabolites detected in the three distinct flower groups, we drew a putative ABP of *P. limprichtii*.

The expression patterns of anthocyanin regulatory genes, including R2R3-MYB, bHLH, and WD40 were also investigated (Figure 5b–d). Phylogenetic analysis (Figure 7) shown that PlMYB13 (unigene0062421) and PlMYB4 (unigene0039181) were clustered with *AtMYB75*, *AtMYB90*, and *AtMYB113*, which belong to subgroup 6 of *A. thaliana* [27], and have been demonstrated to activate anthocyanin accumulation, while *PlMYB1*0 (Unigene0058559) was homologous to *AtMYB11*, *AtMYB12*, and *AtMYB111*, which belong to subgroup 7 in A. thaliana, and have been suggested to control flavonol biosynthesis [28]. The expression pattern of PlMYB13 was consistent with anthocyanin accumulation, while *PlMYB10* exhibited an inverse relationship between its expression and flower color intensity.

### 2.5. The Relationship between Structure Genes and TFs

Through the analysis of the expression patterns of ABP-related genes and the phylogenetic tree of R2R3-MYB, we obtained several candidate genes that correlate with floral color intensity. We constructed a co-expression network (Figure 8) to identify the interactions between ABP-related genes and MBW complex proteins (MYB, bHLH, and WDR). The results show that ten structure unigenes and five MBW complex transcriptional unigenes composed of two R2R3-MYB, two bHLH and one WD40 unigene exhibited interaction relationships. The expression pattern of *PlMYB10* coincided with those of PlbHLH20 (Unigene0062784) and PlbHLH26 (Unigene00660), while *PlbHLH20* and *PlbHLH26* also correlated with *PlWD40-1* (Unigene0002153). PlMYB10 also showed a linear relationship with *PlFLS*. We therefore inferred that *PlMYB10* may interact with *PlbHLH20* or *PlbHLH26*, and *PlWD40-1* to form a MBW transcriptional complex, and regulate the expression pattern of *PlFLS*, finally affecting flower polymorphism in *P. limprichtii.*

## 3. Discussion

We investigated the distribution of color polymorphic individuals in three rock populations of *P. limprichtii* within Huanglong District and combined chemical detection and transcriptomic analysis to isolate the main pigment compounds and candidate genes that determine flower color intensity. We present a putative biosynthesis pathway and discuss the regulatory mechanisms of color formation.

To explore color variation formation factors, we used the CIELAB evaluation system to distinguish rose-purple, pink, and white flower color, and then counted the number of individuals of each of these phenotypes in each rock population. In all three populations the color distribution pattern was nearly consistent, with a color ratio of rose-purple 6:pink 3:white 1. This distribution pattern is very rare in natural color polymorphic populations. Studies have shown that intraspecific flower color variation is often attributed to genetic drift, pollination-mediated selection, environmental conditions, or herbivory [13,29,30,31]. In our experiments, the populations grow across a small range, with the whole population occurring on similar rocks and exposed to the same climatic conditions. Environmental elements such as temperature, drought stress, and ultraviolet radiation were therefore not considered to be crucial promoters of color variation. Pollinator-mediated selection plays an important role in color variation, especially for deceptive pollination species in which competition for pollinators in sympatry promotes flower color divergence [32], and shifts in pollinators also contribute to the macro-evolution of flowers color [22]. Thus, we inferred that the flower color polymorphism within these populations might have been induced by pollinators. Color polymorphism may be a consequence of pollination competition or specific adaptations to pollinators, and pollinator behavior exerts strong selection stress on color variations. Some research also showed adaptive selection for pigmented flowers because colored flowers are less likely to be disrupted by herbivories than colorless ones [3]. According to our field observations, we found that white-flowered individuals were more susceptible to damage than individuals with pigmented flowers, and that white flower petals and cores were severely foraged when blooming. We thus inferred that the dominant pigmented color was beneficial to avoid herbivory, and reduce damage by herbivores to the population. When individuals were damaged they also suffered reduced attractiveness to pollinators, which is not conductive to the stability and development of the population. This explains the distribution pattern of the number of rose-purple flowers in the population. There is another view that flower color may not be the primary goal of natural selection, nor the initial choice of pollinators. Indeed, the biosynthetic precursors of pigments not only display color variations, but also serve other physiological functions [14]. Studies have shown that secondary metabolites associated with plant defense functions share the same biosynthetic pathway, the flavonoid synthesis pathway, correlating pigment with defense ability [33]. Therefore, colorful-flowered individuals were more resistant to some adversities. In summary, the phenomenon that rose-purple flowers were frequent and white ones rare within the population may be mediated by the pollinators and herbivores, and also related to survival adaptability of the *P. limprichtii*. The 6:3:1 distribution pattern of color polymorphism might be a reproductive strategy for the population to maintain the maximum population density, but further evidences should be investigated.

In theory, the transition from pigmented to white flowers could involve any mutations that block one or more steps in the anthocyanin pathway. This includes loss-of-function mutations in any pathway enzyme-coding genes, as well as the cis-regulatory genes that influence any of the pathway enzymes [34,35]. In our study, expression analysis identified several obvious differentially expressed genes in the petal which were down-regulated in white samples compared to pigmented samples, but metabolite detection found that Cy- and Del-derivatives existed in both white petals and pigmented petals, indicating that the color variations, especially the white petals, do not lacking any steps of the anthocyanin pathway. The cis-regulation of transcription factors is a crucial element to promote color divergence. This result is similar with the white color formation in *Primula vulgaris* which is caused by different genes expression pattern rather than loss- of-function mutations leading to the lack of anthocyanin [36].

For ABP-related genes, we isolated 21 transcripts which encode ten enzymes. Seven of the ten were flavonoid synthase genes, including *PlCHS*, *PlCHI*, *PlF3H*, *PlF3′H*, *PlF3′5′H*, *PlDFR*, and *PlANS*; one was a proanthocyanidin synthase gene, *PlANR*; one was a flavonol synthase gene, *PlFLS*; and one was an anthocyanin synthase gene, *PlUFGT*. Most of them are multi-gene families, only *PlF3H*, *PlF3′5′H*, *PlFLS*, and *PlANR* are single copy. To clearly illustrate the catalyzation steps of the ABP, we regarded Cy-related and Del-related biosynthesis processes as independent branches [8]. Thus we did not have to measure the content of each anthocyanin compound. Each branch of anthocyanin synthase was considered to make an equivalent contribution to pigmentation. Considering that both the Cy-related and Del-related branches may share the majority of enzymes, here, we selected ABP-related genes on the Cy-related branches to analyze their expression patterns.

The expression analysis of early step structural genes revealed a high level of *PlCHS* and *PlCHI* expression in white petals and a low level in pigmented petals, suggesting that white petals can produce a large amount of naringenin but cannot eventually flux this into anthocyanin synthase. Meanwhile, the expression levels of *PlF3′H*, *PlF3′5′H*, *PlDFR*, *PlANS*, and *PlUFGT* in pigmented flower petals was high, compared with white ones, and we inferred that downstream structural genes make a large contribution to coloration. Analysis also showed that *PlFLS* was significantly more upregulated in white petals than in pigmented petals. *FLS* encoding enzymes lead substrate into the flavones and flavonols pathway [37]. It has been suggested that the competition between the anthocyanin synthesis pathway and the flavone and flavonols pathways mainly results in substrate competition between *FLS* with *DFR*, while the *FLS* enzyme strengthens the metabolic flux toward the flavonols and limits anthocyanin accumulation [38]. This situation has also been reported in other species, such as in *Paeonia ostii*, a higher expression of *PoFLS4* in the nearly white flowers promotes dihydroflavonols transition into flavonols [11]. In onions, enhanced *AcFLS* could maximize flavonol production in the sheath [39]. Finally, in *Muscari armeniacun*, the conversion of substrate between *FLS* and *DFR* facilitates the elimination of blue pigmentation [8]. Thus, we confirmed *PlFLS* as one of the candidate genes for white color formation in *P. limprichtii*. The up-regulation of *PlDFR* in pigmented flowers is closely accompanied by a decrease of *PlFLS*; hence, more dihydroflavonols flow into anthocyanin production in pigmented flowers. From our analysis, the expression patterns of *PlANS* and *PlUFGT* are correlated with color intensity, and they showed their highest expression levels in the rose-purple flowers and their lowest in the white flowers. The ANS (Anthocyanidin synthase) encoding enzyme catalyzes the conversion of colorless leucocyanidin into colored anthocyanin [40], and anthocyanin is further glycosylated by different UFGT (UDP flavonoid glucosyl transferase) encoding enzymes that convert the anthocyanidins to different anthocyanin derivatives, exhibiting the final color [41]. We therefore speculate that *PlANS* and *PlUFGT* are two crucial genes that determine color intensity in *P. limprichtii*.

It has been revealed that the MBW protein complex, a combination of R2R3-MYB and bHLH transcription factors, along with WD40 proteins, play an important role in regulating the transcription of structural genes [41,42,43,44,45]. The activities of R2R3-MYB factors have distinct roles in determining the action of the complexes, either to promote or inhibit the transcription of anthocyanin biosynthesis genes [46,47]. By combining R2R3-MYB phylogenetic and co-expression network analyses, we isolated *PlMYB10*, which was homologous with the *AtMYB11*, *AtMYB12*, and *AtMYB111* belonging to S7 in *Arabidopsis* and that have been demonstrated to contribute to the regulation of genes that account for anthocyanin accumulation in all tissues [27,28]. The expression pattern was high expression in nearly white flower petals, gradually reducing in flowers with increasing color intensity. It also corresponded with the color polymorphism phenotypes. In other species, overexpression of *AmMYB330*, a negative regulator of the flavonoid biosynthesis, has been proven to inhibit phenylpropanoid metabolism in transgenic tobacco (*Nicotiana tabacum*) plants [48]. The co-expression network showed that *PlbHLH20* and *PlbHLH26*, along with *PlWD40-1* have a strong relationship to *PlMYB10*. It is likely that this potential MBW complex *PlMYB10*/*PlbHLH20*/*PlWD40-1* or *PlMYB10*/*PlbHLH26*/*PlWD40-1* may serve as a repressor responsible for the variation in color intensity in *P. limprichtii*. Co-expression network revealed *PlFLS* is the most likely target gene interacting with *PlMYB10.* Previous studies have verified that R2R3-MYB can regulate the expression pattern of *FLS* through the overexpression of *PsMYB114L* (from *Paeonia suffruticosa*) in *Arabidopsis* [49], which is consistent with our results. Thus, we tentatively speculate about the ABP of *P. limprichtii* (Figure 9). Further studies examining sequencing differences in these candidate genes and *PlMYB10* are necessary to assess our speculations. Metabolic substance quantification is also necessary to confirm the leading pigments in *P. limprichtii.*

## 4. Materials and Methods

### 4.1. Plant Materials

*Pleione limprichtii*, belongs to the Orchidaceae family, and is distributed in the Huanglong Nature Reverse in Sichuan Province, China, with three distinct color variations including rose-purple, pink, and white (Figure 10a–c). The study site was located at 32.68°N, 104.04°E, and 1851.61 m above sea level (Figure 10d). The plants are generally found living on rocks along streams. The sexual reproduction of *Pleione* is commonly dependent on deceptive pollination and each individual usually grows only one flower. The majority of individuals in the population have rose-purple flowers, while white flowers are relatively less common.

### 4.2. Quantitative Statistics and Flower Colorimeter Analysis

Three rocks with *P. limprichtii* populations which contained individuals with all three color variations were selected, the number of each color individuals were counted. In addition, in order to evaluate the flower color objectively, a hand spectrophotometer (CS-280, Hangzhou Color Spectrum Technology Co., Ltd., Hangzhou, China) was used to measure the CIE L*a*b* color components with five technical repetitions using five different parts of the selected petals, in order to evaluate the flower color objectively. The formula C* = $\sqrt{a{*}^{2}\;+\;b{*}^{2})}$, the L* (lightness) and C* (chroma) through the petals were measured using the lightness coefficient ‘a*’, which indicates greenness to redness as the value increases from negative to positive, and ‘b*’, which represents blueness to yellowness. Principle component analysis was performed using the a* and b* values as the principle components [32].

### 4.3. Sampling and RNA Extraction and cDNA Synthesis

Petals from rose-purple, pink, and white flowers at the full bloom stage were sampled, immediately frozen in liquid nitrogen, and preserved at −80 °C before pigment analysis and RNA extraction. RNA was isolated using a Quick RNA isolation Kit (Huayueyang Co., Ltd., Beijing, China), according to the manufacturer’s instructions. The concentration and purity of the RNA was measured with a NanoDrop 2000 (Thermo Fisher Scientific Co., Ltd., Waltham, MA, USA) and Agilent 2100 (Agilent Technologies Co., Ltd., Palo Alto, CA, USA) to verify RNA integrity. A total of nine samples, including three biological replicates, with high concentrations of RNA for each of the three color morphs were selected, then the strand cDNA synthesis was performed using a Revert Aid First Strand cDNA Synthesis Kit (Thermo Fisher, Foster City, CA, USA), according to the manufacturer’s instructions, and was stored at −80 °C for RT-qPCR assays.

### 4.4. Measurement of Flower Anthocyanin

Sample preparation and extraction methods were as follow, the freeze dried tissues were crushed using a mixer mill (MM 400, Verder Shanghai Instruments and Equipment Co., Ltd., Shanghai, China) with a zirconia bead for 1.5 min at 30 Hz. A 100 mg sample of the powder was weighed and extracted overnight at 4 °C with 1.0 mL 70% aqueous methanol. Following centrifugation at 10, 000× *g* for 10 min, the extracts were absorbed and filtered. Then the sample extracts were analyzed using a UPLC system, the analytical conditions were as follow, HPLC: column, Waters ACQUITY UPLC HSS T3 C18 (1.8 µm, 2.1 mm × 100 mm); solvent system, water (0.04% acetic acid): acetonitrile (0.04% acetic acid); gradient program, 95:5 *v*/*v* at 0 min, 5:95 *v*/*v* at 11.0 min, 5:95 *v*/*v* at 12.0 min, 95:5 *v*/*v* at 12.1 min, 95:5 *v*/*v* at 15.0 min; flow rate,. 0.40 mL/min; temperature, 40 °C; injection volume: 2 μL [32,50]. The mass spectrometry data was analyzed using the software Analyst 1.6.1 (AB Sciex Pte. Ltd., Concord, Ontario, Canada) and based on the Metware Database (MWDB, Metware Biotechnology Co., Ltd., Wuhan, China), a local self-built database, and a public metabolite information database, to identify anthocyanin compounds. The experiments were repeated three times. 1.8 µm

### 4.5. Library Preparation and Sequencing

The construction of the libraries and RNA-seq were performed by the Omicshare Biotechnology Corporation (Guangzhou, China). The mRNA was enriched with Oligo (dT) beads, then broken into short fragments and reverse transcribed into cDNA as templates. First and second-strand cDNA were then synthesized. The cDNA fragments were purified with a QiaQuick PCR extraction kit (Qiagen, Valencia, CA, USA) and end repaired. Poly (A) was added and ligated to Illumina sequencing adapters. They were then sequenced using the Illumina HiSeqTM 4000 (Illumina Inc., Centre Drive, San Diego, CA, USA).

### 4.6. De Novo Transcriptome Assembly Annotation

Transcriptome de novo assembly was performed with clean data, filtered from the raw data by removing adaptors and unknown nucleotides (>10%), and those with low quality reads. The data were assembled using the Trinity platform [51] with the parameters ‘K-mer = 31′ and ‘K-mer cover = 6′. First, short reads of a certain length were combined with overlap to form longer contigs. Then, based on their paired-end information, clean reads were mapped back to the corresponding contigs. Thus the sequences of the transcripts were finished, and defined as unigenes. All assembled unigenes were then annotated using BLASTx (E-value ≤ 1 × 10^-5^) against protein databases, including the National Center for Biotechnology Information non-redundant (Nr, ftp://ftp.ncbi.nih.gov/blast/db/), Swiss-Protein (https://www.uniprot.org/), Kyoto Encyclopedia of Genes and Genomes (KEGG, https://www.genome.jp/kegg/), and Gene Ontology (GO, http://geneontology.org/) databases. while Nr and Swiss-Protein annotate gene function, KEGG is used to understand biological systems and GO divides genes into different categories. BLASTx was used to search for the unigenes against the public databases with the following order of priority: Nr, Swiss-protein, KEGG, and COG. When a unigene could not be aligned to any of these protein databases, the protein code sequence and sequence direction was confirmed using the ESTscan program.

### 4.7. Expression Profile and RT-qPCR

To compare color-related unigenes expression divergence, they were first normalized to RPKM (reads per kb per million reads). After that, different sample ratios of RPKM values were calculated. The FDR (false discovery rate) value was used to identify the threshold of the *p*-value in multiple tests in order to compute the significance of the differences among unigenes. Here, only FDR significance scores < 0.05, and log2 ratios > 1 were regarded as differentially expressed genes and used in subsequent analysis. In order to validate the expression pattern of the RNA-Seq results, ten important different expression genes (the primers used are listed in Appendix A) were selected and measured using RT-qPCR on a Quant Studio 5 Real-Time PCR System (Thermo Fisher, Foster City, CA, USA) using the PowerUp^TM^ SYBR^TM^ Green Master Mix (Thermo Fisher, Foster City, CA, USA), according to the manufacturer’s instructions. *PlUBC34* and *PlUBC3*5 (ubiquitin-conjugating enzyme; primers are also listed in Appendix A) actin genes from *Pleione* were used as the internal control for the normalization of gene expression. Each sample (including three biological repetitions) was quantified in triplicate.

### 4.8. Genes Related to the ABP and Phylogenetic Analyses

The ABP-related structure genes and transcription factors including *CHS*, *CHI*, *F3H*, *F3′5′H*, *DFR*, *FLS*, *ANS*, *FNS*, *UFGT*, *MYB*, *bHLH*, and *WD40* were used as queries to retrieve the corresponding unigenes from the *P. limprichtii* libraries. Meanwhile, every reference gene from model plants obtained from KEGG were aligned with the libraries using BLASTx to search for more related unigenes.

For the MYB genes, to determine which of the genes belonged to which R2R3-MYB subfamily, the R2R3-MYB genes from *Arabidopsis thaliana* were used to conduct phylogenetic analyses (Amino acid sequence obtained from Gene Bank, accession number is listed in Appendix A), and MEGA7.0 software (Institute of Molecular Evolutionary Genetics, PA, USA) [52] was used to perform sequence aligning and construct a circular phylogenetic tree according to neighbor joining method with 1000 interactions.

### 4.9. Structure Genes and Tfs Co-Expression Network

The String online database (https://string-db.org/) was used to search for the interaction relationships between structure genes and transcription factors. Then we used Cytoscape software (National Institute of General Medical Sciences, Bethesda, MD, USA) to construct a co-expression network to identify their main impact factors [53].

## 5. Conclusions

The ratio of 6:3:1 distribution patterns of *P. limprichtii* within the population in Huanglong District seems to reveal that pollinators, herbivores, and survival adaptability could promote the development of such a reproductive strategy in the population to maintain the maximum population density. Function and expression patterns indicated that *PlFLS* probably is a crucial gene in the formation of white and pigmented flowers. *PlANS* and *PlUFGT* were found to determine the color intensity in pigmented flowers. In addition, a putative MBW complex, *PlMYB10*/*PlbHLH2*0/*PlWD40-1* or *PlMYB10*/*PlbHLH26*/*PlWD40-1* may serve as a repressor of regulated *PlFLS* expression that is responsible for variation in the color intensity of *P. limprichtii*. Our results provide valuable molecular information on floral color variations in *Pleione*, and also provide inspiration to further explore the relationship between color polymorphism and species evolution and to study its contribution to color evolution.

## Figures and Tables

**Figure 1 ijms-21-00247-f001:**
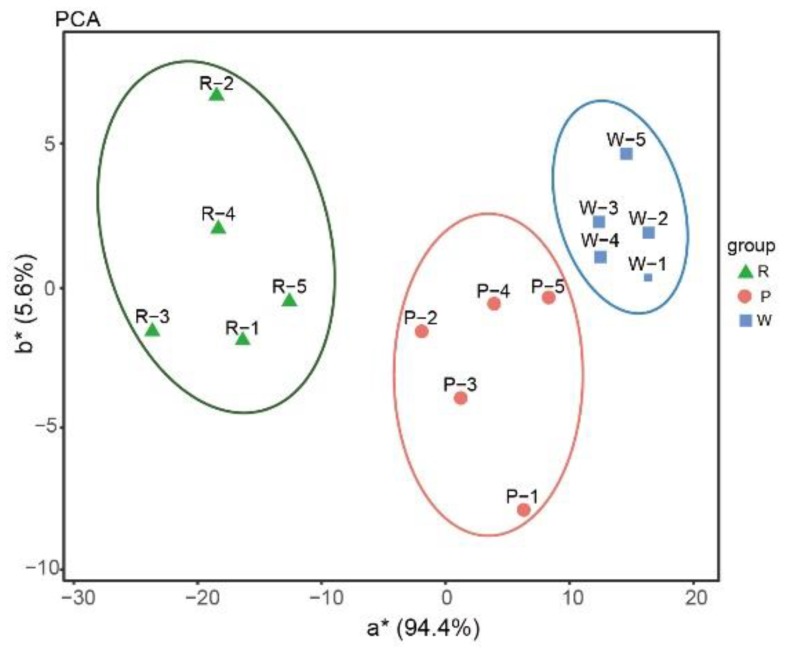
Principal component analysis of petal color of *P. limprichtii*. Distribution based on bivariate values of a* (redness and greenness) and b* (yellowness and blueness). a* means difference in red and green, b* means difference in yellow and blue.

**Figure 2 ijms-21-00247-f002:**
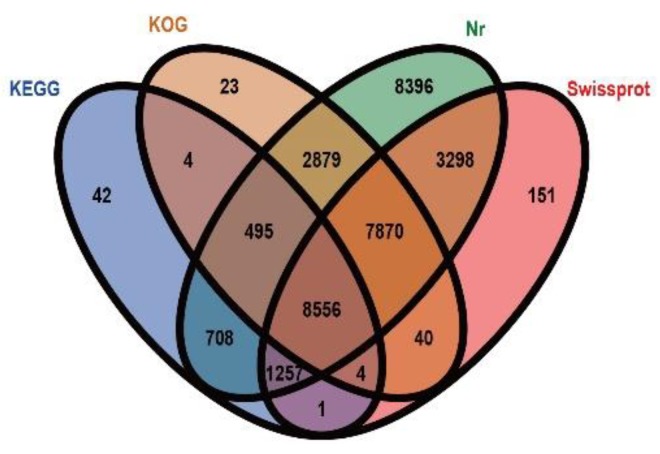
Venn diagram of the number of unigenes annotated by BLASTx with four protein databases.

**Figure 3 ijms-21-00247-f003:**
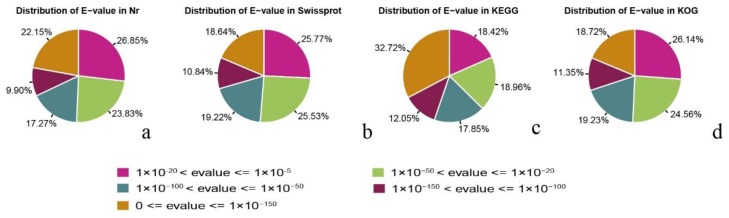
E-value distribution of top BLASTx hits against four protein databases for each unigene. (**a**) distribution E-values in non-redundant protein database; (**b**) distribution E-values in Swiss-protein database; (**c**) distribution E-values in the Kyoto Encyclopedia of Genes and Genomes database; (**d**) distribution E-values in the Eukaryotic Orthologous Group database.

**Figure 4 ijms-21-00247-f004:**
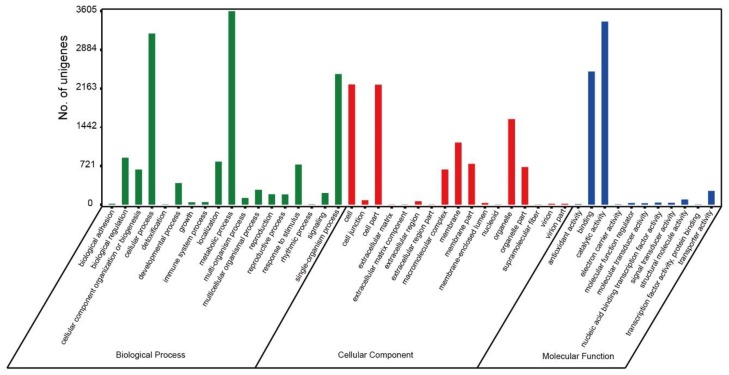
Function classification of the gene ontology of all unigenes based on Nr annotation.

**Figure 5 ijms-21-00247-f005:**
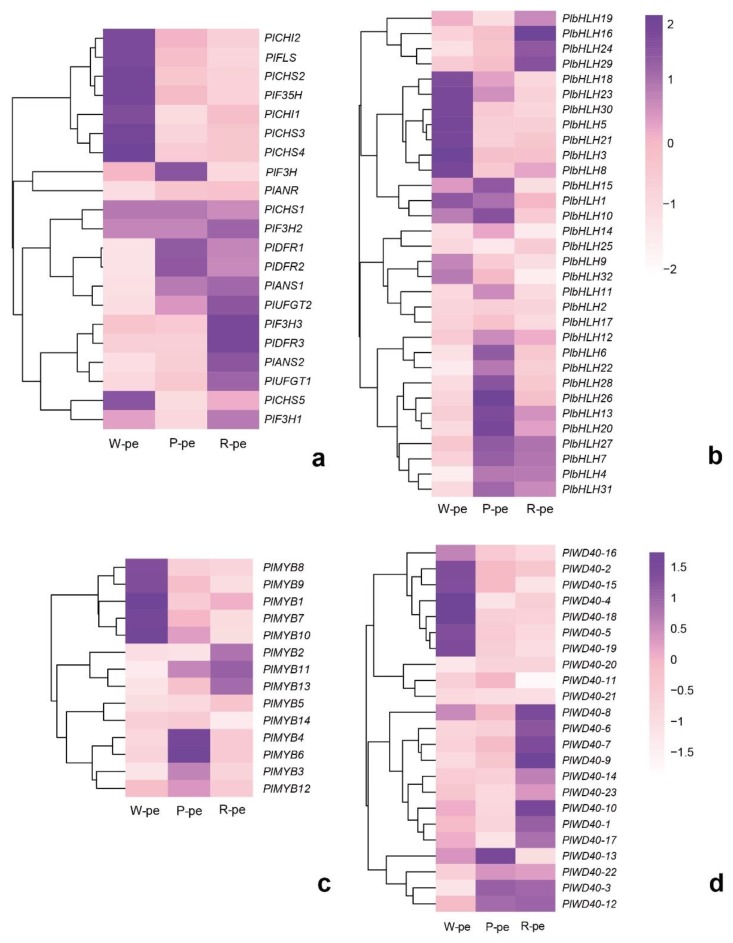
Expression pattern analysis base on RNA sequencing of anthocyanin biosynthesis pathway-related genes and transcription factors in *P. limprichtii*. (**a**) Anthocyanin biosynthesis pathway-related unigenes; (**b**) *bHLH* unigenes; (**c**) *R2R3-MYB* unigenes; (**d**) *WD40* unigenes. W-pe (petal of white flower), P-petal (petal of pink flower), R-petal (petal of rose -purple flower).

**Figure 6 ijms-21-00247-f006:**
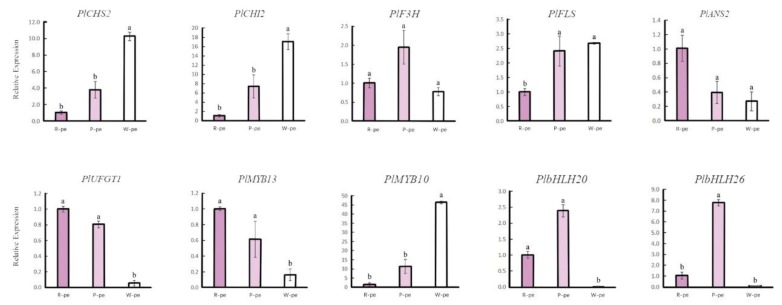
Real time quantitative reverse transcription-PCR of several genes in *P. limprichtii*. Each value is shown as average ± standard deviation from three biological replicate sampling.

**Figure 7 ijms-21-00247-f007:**
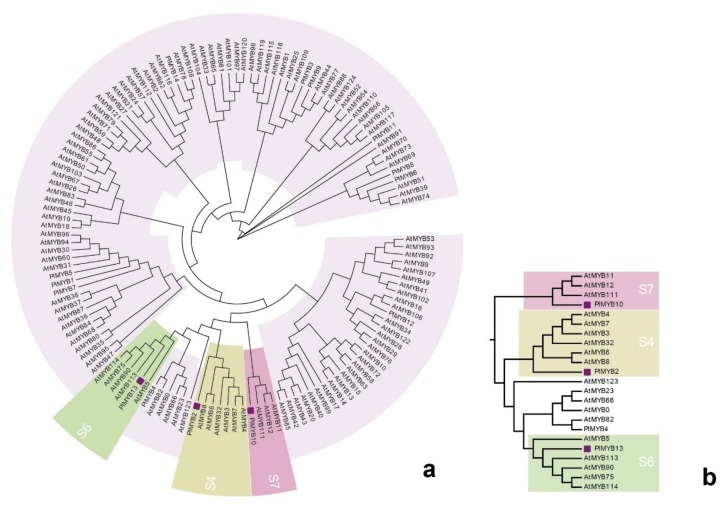
Phylogenetic analysis of R2R3-MYB DNA binding domains for *P. limprichtii* and Arabidopsis thaliana. (**a**) Circular phylogenetic tree; (**b**) amplification of S4, S6, and S7 branches. The R2R3 domains of the 14 MYBs identified in *P. limprichtii* petal transcriptome were aligned and analyzed using neighbor-joining phylogenetic methods.

**Figure 8 ijms-21-00247-f008:**
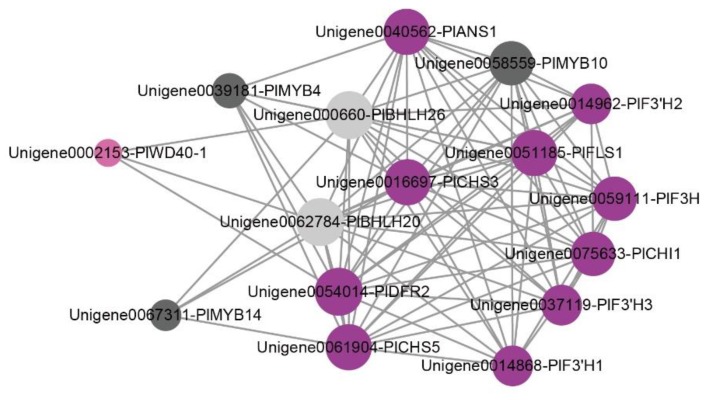
A co-expression network of anthocyanin biosynthesis pathway-related genes and transcription factors involved in pigmentation. Rose-purple circle represents structural genes; dark-gray circle represents R2R3-MYB; light-gray circle represents bHLH; pink circle represents WD40.

**Figure 9 ijms-21-00247-f009:**
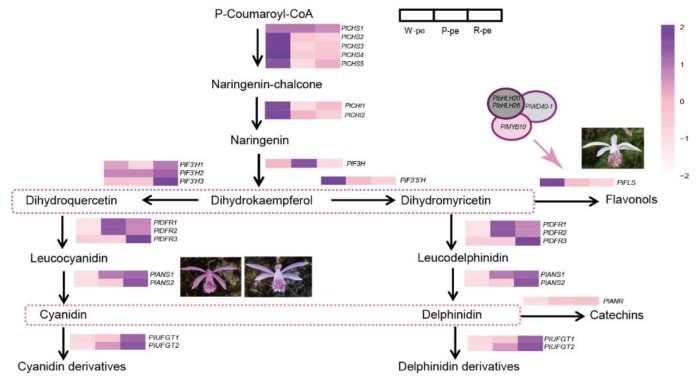
Tentative pathways for *P. limprichtii* color variations. The colored bar is the value of log2 (RPKM + 1), represented using the depth of color, with purple representing the up-regulated expression genes and pink representing the down-regulated expression genes. RPKM means the reads per kb per million reads mapped. CHS, chalcone synthase; CHI, chalcone-flavanone isomerase; F3H, flavanone-3-hydroxylase; F3’H, flavonoid 3’-hydroxylase; F3’5’H, flavonoid 3’5’-hydroxyla; DFR, dihydroflavonols 4-reductase; ANS, anthocyanidin synthase; UFGT, UDP flavonoid glucosyl transferase; FLS, flavonol synthase; The three gene complex consist of a MYB, bHLH and WD40 in most angiosperm.

**Figure 10 ijms-21-00247-f010:**
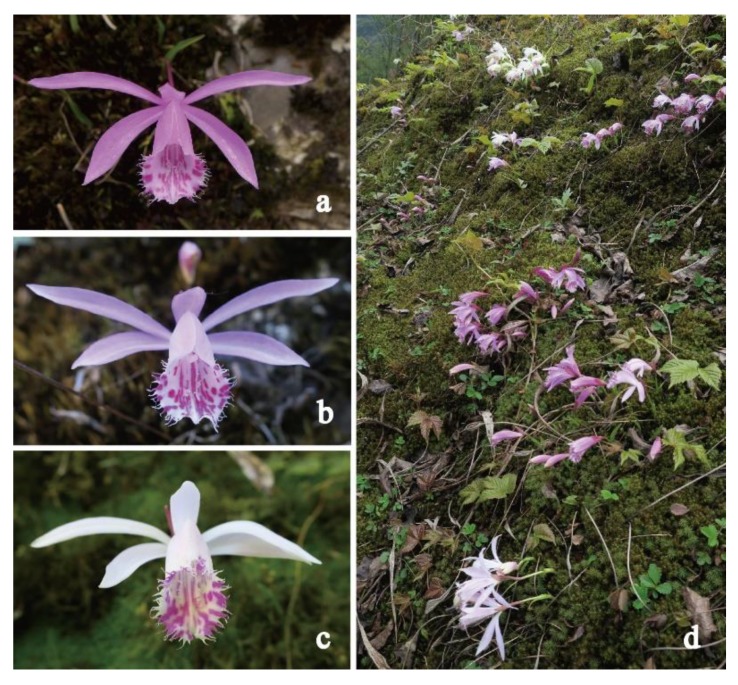
Flower polymorphism of *Pleione* limprichtii in Huanglong population. (**a**) rose-purple flower; (**b**) pink flower; (**c**) white flower; (**d**) one of the polymorphic populations on a rock.

**Table 1 ijms-21-00247-t001:** Number of *P. limprichtii* individuals of each flower color polymorph in three selected populations at the opening stage.

Population	Rose-Purple	Pink	White
population1	409 (58%)	227 (32%)	70 (10%)
population2	31 (65%)	15 (31%)	2 (4%)
population3	92 (60%)	54 (35%)	8 (5%)

The percentage indicates the proportion of individuals with a given color account for the total number of individuals in the population.

**Table 2 ijms-21-00247-t002:** Color parameters of three distinct colors of *P. limprichtii* petal.

Flower Color	RHS	L*	C*	b*/a*
purple red	Purple-group N78B	67.92 ± 10.27	49.64 ± 9.41	−1.68 ± 0.61
		76.66 ± 4.51	52.88 ± 4.18	−2.35 ± 0.23
		67.35 ± 10.34	57.13 ± 6.24	−1.68 ± 0.61
		81.31 ± 8.72	51.89 ± 7.42	−1.91 ± 0.44
		79.68 ± 15.55	46.28 ± 11.76	−1.85 ± 0.79
Pink	Purple-group 75A	61.08 ± 5.29	26.71 ± 1.82	−1.20 ± 0.36
		66.05 ± 6.63	35.45 ± 9.60	−1.77 ± 0.71
		66.35 ± 2.73	31.72 ± 3.14	−1.57 ± 0.29
		58.80 ± 5.20	29.66 ± 6.05	−2.20 ± 1.07
		62.80 ± 5.78	25.43 ± 4.99	−2.56 ± 0.89
White	Purple-group 76D	93.08 ± 5.57	13.31 ± 9.67	−4.04 ± 2.06
		88.12 ± 8.13	18.53 ± 7.31	−13.60 ± 5.70
		95.85 ± 4.48	22.41 ± 6.77	−7.97 ± 3.88
		87.52 ± 11.45	25.27 ± 10.21	−6.11 ± 2.61
		87.07 ± 7.97	21.36 ± 6.12	−9.28 ± 4.73

RHS, Royal horticulture society color chart evaluation index; L*, lightness; a*, b*, chromatic components; C*, chroma (brightness).

**Table 3 ijms-21-00247-t003:** Anthocyanin content of petal in *P. limprichtii*.

Index	Ion Mode	Molecular Weight	Substance
pme3609	positive	287.24	Cyanidin
pme0442	positive	303.24	Delphinidin
pmb0541	positive	697.1	Cyanidin 3-O-glucosyl-malonylglucoside
pmb0542	positive	535.1	Cyanidin 3-O-malonylhexoside

**Table 4 ijms-21-00247-t004:** Tentative anthocyanin biosynthesis pathway-related genes in *P. limprichtii*.

Pathway	Gene	Encoding Enzyme	Number
Flavonoid biosynthesis	*CHS*	Chalcone synthase	5
	*CHI*	Chalcone isomerase	2
	*F3H*	Flavanone 3-dioxygenase	1
	*F3′H*	Flavonoid 3’-hydroxylase	3
	*F3′5′H*	Flavonoid 3’,5’-hydroxylase	1
	*DFR*	Dihydroflavonol 4-reductase	3
	*ANS*	Anthocyanidin synthase	2
	*ANR*	Anthocyanidin reductase	1
	*FLS*	Flavonol synthase	1
Anthocyanin biosynthesis	*UFGT*	Anthocyanidin 3-O-glucosyltransferase	2

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
