# Peer review of "Comparative Transcriptomics Provides Insight into Floral Color Polymorphism in a Pleione limprichtii Orchid Population"

_ijms, 2019, doi:10.3390/ijms21010247_

Round 1

Reviewer 1 Report

This is a very interesting and well written manuscript, about the pigment distribution and the floral color formation in Pleione limprichtii, an important topic for theoretical evolutionary reasons, as well as functional genetics. I have not found anything that needs improvement in this manuscript, besides the possibility to improve english language, especially as far as syntax is concerned. Otherwise, there are some minor remarks that follow, mainly in the abstract and the introduction.

- line 22: The sentence "We inferred that this is a reproductive strategy to maintain the population size" is a little difficult to understand and should be rewritten.
- line 35: Pleione should be in italics (genus).
- line 45: What are "sister populations"?
- Line 85: Is the sentence "We propose that white mutants occur through blockage of the ABP, and focus on revealing the coloring patterns and formation mechanisms of flower color polymorphism within the population" a finding of this study, or a hypothesis to be tested? If it is the first case, then it belongs in the discussion. If it is a hypothesis, then it should be stated clearly.
- Line 87: The last sentence of the introduction is a nice conclusion that fits at the end of the manuscript and not at the end of the introduction.
-Line 258: I suggest to replace the word "dominant" with "frequent", as dominant can be understood in a Mendelian way (which is not the case here).

Author Response

Question 1: Line 22: The sentence "We inferred that this is a reproductive strategy to maintain the population size" is a little difficult to understand and should be rewritten.

Answer: Thanks for you good suggestion. The sentences had been revised as “We inferred that the distribution pattern may serve as a reproductive strategy to maintain the population size.” to highlight the distribution pattern is an important reproductive strategy under the same environment. (see the line 22)

Question 2: Line 35: Pleione should be in italics (genus).

Answer: Thank you for your question. I am sorry for a low-level error occurred. I have changed "Pleione" to "P. limprichtii". (see the line 35)

Question 3: Line 45: What are "sister populations"?

Answer: Thank you for your question. I have changed the "sister populations" to "sister species" following the cited reference No. 7. (see the line 45)

Question 4: Line 85: Is the sentence "We propose that white mutants occur through blockage of the ABP, and focus on revealing the coloring patterns and formation mechanisms of flower color polymorphism within the population" a finding of this study, or a hypothesis to be tested? If it is the first case, then it belongs in the discussion. If it is a hypothesis, then it should be stated clearly.

Answer: Thank you for your kind advice. I just deleted it since the sentence with the same meaning already existed in the Discussion Section. (see Line 265-272)

Question 5: Line 87: The last sentence of the introduction is a nice conclusion that fits at the end of the manuscript and not at the end of the introduction.

Answer: Thank you for your suggestion. Since the sentence has the same meaning with the last one of our manuscript, I just deleted it from the end of introduction. (see the line 93)

Question 6: Line 258: I suggest to replace the word "dominant" with "frequent", as dominant can be understood in a Mendelian way (which is not the case here).

Answer: Thank you for your advice. I have revised "dominant" to "frequent". (see Line 256)

Reviewer 2 Report

In this manuscript, the authors selected the P. limprichtii population in Huang-Long, Sichuan Province, China, which displayed three color variations: rose-purple, pink, and white, providing ideal material to explore color variations with regard to species evolution. The authors investigated the distribution pattern of the different color morphs. The ratio of rose-purple: pink: white-flowered individuals was close to 6: 3: 1. The authors also inferred that this is a reproductive strategy to maintain the population size. Metabolome analysis was used to reveal that cyanindin derivatives and delphidin are the main color pigments involved. RNA sequencing was used to characterize anthocyanin biosynthetic pathway-related genes and reveal different color formation pathways and transcription factors in order to identify differentially expressed genes and explore their relationship with color formation. In addition, qRT-PCR was used to validate the expression patterns of some of the genes. The results show that PlFLS serves as a crucial gene that contributes to white color formation and that PlANS and PlUFGT are related to the accumulation of anthocyanin which is responsible for color intensity, especially in pigmented flowers. Phylogenetic and co-expression analyses also identified a R2R3-MYB gene PlMYB10.

In general, the study needs more improvement before considering its publication based on the following major points;

- Introduction:

The introduction has some English mistakes and grammar errors, so the authors should revise that.

A section of the research hypothesis should be added to more clarify the question and importance of this study.

Some sentences were written without citing them. Please cite those sentences.

Recent references should be also included.

- Material and methods:

More information should be added regarding the description of the methods used in this study.

Statistical analysis should be explained in details

- Results:

The results showed some significant data. However, the resolution of some figures is bad and those figures should be replaced with higher solution ones such as Figures 2, 4, 5, 6 and 7.

- Discussion:

The discussion should be merged together (not separately written).

The results should be repeated. The interpretation of results and discussion is recommended.

The results should be compared with the literature findings

- Conclusion: 

Conclusions should be revised and include only the significant findings arisen from this manuscript.

- References:

References should be formatted as the authorship guidelines.

Recent references should be added.

Author Response

Question 1: The introduction has some English mistakes and grammar errors, so the authors should revise that.

Answer: Thank you for your advice and patience. I am so sorry for taking you so much time to read. The manuscript has been corrected by the English editing service at Editage.

Question 2: A section of the research hypothesis should be added to more clarify the question and importance of this study.

Answer: Thank you for your suggestion. I have added a paragraph to clarify the research hypothesis. "The focus of our study was to understand the molecular mechanism of color polymorphism, including how the white individuals formed and the main reason caused pink flowers intensities, as well as summarized the general rules of the color distribution pattern. We aimed to investigated distribution of color monomorphic in the Huang-Long P. limprichtii population and examined the transcriptome and biochemistry of their color polymorphic petals. RNA sequencing (RNA-seq) and ultra-performance liquid chromatography (UPLC) were used to identify the variation of related genes and the differences in flavonoid intermediates in the ABP that cause color transition, respectively." (see Line 86-93)

Question 3: Some sentences were written without citing them. Please cite those sentences.

Answer: Thank you for your valuable suggestion. I have annotated references in the text. Such as the introduction sentences "Due to the instability, anthocyanidins exist mainly as anthocyanins, which are formed by anthocyanidins and various glycosides [16]. They play an irreplaceable role in the color development of plants and are primarily derived from three main anthocyanidins: pelargonidin (brick red to scarlet), cyaniding (red to magenta), and delphinidin (purple to violet) [16]." (see Line 62-65)

Question 4: Recent references should be also included.

Answer: Thank you for your suggestion. I have added some recent references in introduction and discussion, the number is the 4th and 5th in introduction and the 35th, and 36th in discussion. (see Line 43-44; Line 264-271)

Question 5: More information should be added regarding the description of the methods used in this study.

Answer: Thank you for your advice. I have added more details of anthocyanin extraction and qRT-PCR performance. (see Line 366-377; Line 401-412)

Question 6: Statistical analysis should be explained in details.

Answer: Thank you for your advice. I have added the sentence "The percentage indicates the proportion of individuals with a given color account for the total number of individuals in the population." To explain the statistical analysis. (see Line 104-105)

Question 7: The results showed some significant data. However, the resolution of some figures is bad and those figures should be replaced with higher solution ones such as Figures 2, 4, 5, 6 and 7.

Answer: Thank you for your suggestion. I have shown significant data in Figure 7; the resolution of all the picture in this study is 300 dpi which was compressed  in the text. To solve this problem, I will submit the original high in a zip file.

Question 8: The discussion should be merged together (not separately written).

Answer: Thank you for your suggestion. I have merged discussion text into one section. (see Line 221-332)

Question 9: The results should be repeated. The interpretation of results and discussion is recommended.

Answer: Thank you for your suggestion. The results of RNA-seq has been validated by qRT-PCR method, the result is accurate and reliable. Also, I have interpretation of the results and discussion to summarize final conclusion.

Question 10: The results should be compared with the literature findings.

Answer: Thank you for your suggestion. I have compared some of our results with the literature findings, such as the reason of white flower formation with that of Primula vulgaris. "This result is similar with the white color formation in Primula vulgaris, different genes expression pattern rather than loss- of- function mutations leads to the lack of anthocyanin." And the result of PlMYB10, act as a repressor to inhibit the transcription of anthocyanin biosynthesis genes, has been compared to other literature, "In other species, overexpression of AmMYB330, a negative regulator of the flavonoid biosynthesis, has been proven to inhibit phenylpropanoid metabolism in transgenic tobacco (Nicotiana tabacum ‘Samsun NN’) plants." (see Line 270-272; 319-321)

Question 11: Conclusions should be revised and include only the significant findings arisen from this manuscript.

Answer: Thank you for your suggestion. I have revised and shorten the conclusion. "The ratio of 6: 3:1 distribution patterns of P. limprichtii within the population in Huang-Long District seems to reveal that pollinators, herbivores, and survival adaptability could promote the development of such a reproductive strategy in the population to maintain the maximum population density. Function and expression patterns indicated that PlFLS probably is a crucial gene in the formation of white and pigmented flowers. PlANS and PlUFGT were found to determine the color intensity in pigmented flowers. In addition, a putative MBW complex, PlMYB10/ PlbHLH20/ PlWD40-1 or PlMYB10/ PlbHLH26/ PlWD40-1 may serve as a repressor of regulated PlFLS expression that is responsible for variation in the color intensity of P. limprichtii. Our results provide valuable molecular information on floral color variations in Pleione, and also provide inspiration to further explore the relationship between color polymorphism and species evolution and to study its contribution to color evolution."(see Line 429-439)

Question 12: References should be formatted as the authorship guidelines.

Answer: Thank you for your suggestion. I have revised the reference as the authorship guidelines required.

Question 13: Recent references should be added.

Answer: Thank you for your suggestion. See as question 4, I have added some recent references in introduction and discussion. (see Line 43-44; Line 264-271)

Round 2

Reviewer 2 Report

The authors addressed my concerns and greatly improved the paper.